# Crosstalk between Autophagy and Type I Interferon Responses in Innate Antiviral Immunity

**DOI:** 10.3390/v11020132

**Published:** 2019-02-01

**Authors:** Yu Tian, Ming-Li Wang, Jun Zhao

**Affiliations:** 1Department of Microbiology, Anhui Medical University, Hefei 230032, China; tianyu8464@gmail.com; 2Wuhu Interferon Bio-Products Industry Research Institute Co., Ltd., Wuhu 241000, China

**Keywords:** antiviral immunity, type I interferon, autophagy

## Abstract

Autophagy exhibits dual effects during viral infections, promoting the clearance of viral components and activating the immune system to produce antiviral cytokines. However, some viruses impair immune defenses by collaborating with autophagy. Mounting evidence suggests that the interaction between autophagy and innate immunity is critical to understanding the contradictory roles of autophagy. Type I interferon (IFN-I) is a crucial antiviral factor, and studies have indicated that autophagy affects IFN-I responses by regulating IFN-I and its receptors expression. Similarly, IFN-I and interferon-stimulated gene (ISG) products can harness autophagy to regulate antiviral immunity. Crosstalk between autophagy and IFN-I responses could be a vital aspect of the molecular mechanisms involving autophagy in innate antiviral immunity. This review briefly summarizes the approaches by which autophagy regulates antiviral IFN-I responses and highlights the recent advances on the mechanisms by which IFN-I and ISG products employ autophagy against viruses.

## 1. Introduction

During viral infection, viral pathogen-associated molecular patterns (PAMPs) can be detected by host cell pattern recognition receptors (PRRs). PRRs initiate antiviral responses by recruiting signal molecules that promote the generation of IFN-I and other pro-inflammatory cytokines. Subsequently, IFN-I binds to its receptors, inducing the interferon-stimulated genes (ISGs)-dependent transcription of multiple proteins with antiviral effects [1,2]. There are currently nine subtypes of IFN-I, including IFN-α and IFN-β, which are the most common [3]. Antiviral responses mediated by IFN-I can be considered predominant manifestations of innate immunity.

Autophagy is a critical degradation process in all eukaryotes, mediating the elimination of harmful components, the recycling of the cytoplasm, and the supply of nutrients [4]. During the general autophagic process, superfluous and aberrant cytosolic constituents or organelles are enveloped into a double-membrane compartment, which fuses with the lysosome/vacuole, generating the autophagolysosome. Eventually, the autophagolysosome cargo is degraded by hydrolases [5,6,7]. Autophagy, therefore, plays a crucial role in degrading viruses and viral components during viral infection [8,9]. Autophagy has also been shown to play an antiviral mechanism. With their long-term struggle, viruses have also evolved some tactics to subvert autophagy, thereby, increasing their replication: autophagy vesicles are used by the encephalomyocarditis virus (EMCV) as a site for its RNA replication [10]. These data indicate that autophagy aids viruses in some circumstances, and it has, therefore, been characterized as a double-edged sword in the antiviral immunity [11].

Expanding research in this field has generated numerous studies indicating that the antiviral effects of autophagy cannot be merely attributed to degrading viruses. It has been shown that autophagosomes deliver the virus-derived PAMRs to PRRs and initiate the IFN-I responses of antiviral immunity [9,12]. Autophagic processes have also been used and regulated by IFN-I- and IFN-I-induced proteins to eliminate viruses. Therefore, a crosstalk between autophagy and the IFN-I responses is a novel mechanism that may account for the different phenomena that autophagy exerts during its antiviral or proviral roles. In this review, we (1) introduce the autophagy machinery, (2) investigate the role of autophagy in regulating IFN-I expression and the IFN receptor degradation during innate antiviral immunity, and (3) then focus on how IFN-I and ISG products manipulate autophagy to enhance antiviral immune responses.

## 2. The Types and Mechanisms of Autophagy

There are three types of autophagy: chaperone-mediated autophagy (CMA), microautophagy, and macroautophagy. CMA is a selective autophagic process where an intracellular chaperone protein—HSC70 (heat shock cognate protein of 70 kDa)—recognizes the KFERQ motif of its target proteins. This interaction drives the target proteins to lysosomes for degradation, which occurs via the lysosomal-associated membrane protein 2A (LAMP2A) [13,14,15]. During microautophagy, cytosolic contents are engulfed by direct invagination of the lysosomal membrane, converting isolate vesicles into the lysosome [16]. Macroautophagy can be distinguished from the other two types by its differential double-membrane compartment—named autophagosome—which can sequester cytosolic cargo into the lumen and then fuse with the lysosome, causing cargo degradation [7] (Figure 1).

Macroautophagy and microautophagy have been previously considered to lack the capacity for controlling the selective degradation of cytosolic contents. However, emerging evidence suggests otherwise, where depending on the interaction between the cargo and its receptors, these processes can mediate selective autophagy [17,18]. Aberrant cytosolic components may undergo a series of modifications, mainly ubiquitination. Many cargo receptors such as SQSTM1/p62 can accurately recognize tubiquitination signals via their Ub-associated (UBA) domain and harness their linear motif light chain 3(LC3)-interacting region (LIR) to bind LC3 family proteins. These proteins, which are located in the phagophore’s membrane, engage the formation of autophagosomes and ultimately cause cargo degradation [18,19,20,21]. Other common cargo receptors include NBR1, NDP52, OPTN, TAX1BP1, etc. [22,23,24,25]. 

Macroautophagy is the most commonly known “Autophagy”, and in this paper, it is referred to as autophagy. The process of autophagy is driven by a series of autophagy-related gene (ATG) proteins, which were originally identified and termed in yeast [26,27]. More than 40 ATGs have since been identified in yeast, and their respective homologues have been found in mammals [28]. The core process of autophagy can be subdivided into five major phases: initiation, nucleation, elongation, maturation, and degradation, which are tightly regulated by four ATG protein complexes (Figure 1). Generally, the serine/threonine kinase mammalian target of rapamycin (mTOR) inhibits autophagy [29]. mTOR is the core kinase in cellular metabolic homeostasis, possessing two forms: mTOR complex 1 (mTORC1) and mTOR complex 2 (mTORC2). mTORC1 inactivates autophagy by interacting with the ATG1/unc-51-like kinase 1(ULK1) complex (comprising ATG1/ULK1, ATG13, ATG101, and the FAK family kinase-interacting protein of 200 kDa (FIP200)) and by phosphorylating the autophagy initiators ULK1 and ATG13 [30,31,32]. However, mTORC1 activity is restrained by signals including starvation, stress, and pathogen infection, which contribute to the disintegration of the ATG1/ULK1 and mTORC1 complexes, thus initiating autophagy.

Subsequently, the ATG1/ULK1 complex phosphorylates Beclin-1—a key autophagy protein—promoting the Beclin-1 complex and phosphoinositide 3-kinase catalytic subunit type III complex (PI3KC3) formation. The latter is comprised of Beclin-1, vacuolar protein sorting 34 (VPS34), VPS15, and ATG14L. PI3KC3 is an essential regulator autophagy by nucleation and mediates the formation of pre-autophagosomal structures (PASs) [33,34,35].

PAS elongation depends primarily on two ubiquitin-like (UBL) conjugation systems. Under a series of ubiquitylations driven by ubiquitin-activating enzymes ATG3 and ATG7, ATG12 and LC3 generate the ATG12–ATG5–ATG16L complex and liposoluble LC3-II, respectively. Both products anchor to the PAS membrane and play significant roles in facilitating the extension of PAS via recruiting other membrane structure, thereby forming autophagosomes [36,37,38]. 

As a consequence, autophagosomes fuse with lysosomes and degrade the captured cargo. The last phase is controlled by another Beclin-1 complex, which is comprised of Beclin-1, VPS34, VPS15, and UVRAG. Of note, an autophagy inhibitor, Rubicon, can interact with this Beclin-1 complex and antagonize its activity [39,40].

During innate immunity, autophagy forms a unique approach to pathogen, termed xenophagy. Viral infections result in the cellular autophagy process, and in turn, viral components are captured and degraded in the autophagosomes [41]. Importantly, besides digesting virus, xenophagy also promotes the process by which PRRs recognize viral PAMPs and mediates the IFN-I responses [42,43] (Table 1). 

## 3. Autophagy Regulates the Expression of IFN-I

IFN-I expression is controlled by an upstream PRR signaling pathway, and there are three major groups of PRRs in innate antiviral immunity. The first group is the retinoic acid-inducible gene-I (RIG-I)-like receptor (RLR) family, consisting of RIG-I, melanoma differentiation associated gene 5 (MDA5), and the laboratory of genetics and physiology 2 (LGP2). All of which possess the C-terminal regulatory domain and middle DExD/H box helicase domain that can unwind the RNA duplexes. However, only RIG-I and MDA5 have the N-terminal caspase activation and recruitment domain (CARD) which associates with other CARDs of downstream regulatory molecules to deliver RNA virus signals [44,45]. The second group contains multiple members of the Toll-like receptor (TLR) family: TLR3, TLR7, TLR8, and TLR9; all of which are located in the endosomal membrane and recognize double-stranded RNA (dsRNA) (TLR3), single-strand RNA (ssRNA) (TLR7/8), and unmethylated CpG DNA (TLR9). The last group includes several cytosolic DNA sensors such as cGAS (cyclic GMP-AMP synthase), which play a vital role in recognizing DNA viruses [2,46].

After recognizing PAMPs, PRRs activate downstream signals and induce IFN-I responses. Nevertheless, excessive immune activation may cause damage to the body. In order to acquire beneficial results, host cells have developed intricate approaches to balance the expression of IFN-I, and autophagy is required for these regulatory mechanisms (Figure 2).

### 3.1. Autophagy Regulates IFN-I Production via the RLR Signaling Pathway

During transduction of the RLR signaling pathway, PAMPs recognition can induce conformational changes to RIG-I and MAD5—releasing their N-terminal CARD. The exposed CARD is then ubiquitinated and interacts with the mitochondrial antiviral signaling protein (MAVS) CARD for signaling. MAVS recruits downstream signal molecules such as TANK-binding kinase 1 (TBK1), which phosphorylates and activates the IFN regulatory factor (IRF), resulting in IRF dimerization. As a result, IRF dimers translocate to the nucleus, mediating the generation of IFN-I [47,48,49]. Currently, 11 IRFs are known, and IRF3/7 is involved in regulating IFN production [50].

Previous reports have investigated the relationship between autophagy and the RLR signaling pathway regulation, the mechanisms of which are complicated. Generally, signal transduction suppression is related to the degradation of signaling molecules. Two major protein degradation pathways exist in mammalian cells: the ubiquitin–proteasome system and autophagy [51]. Currently, the majority of known RIG-I negative factors, such as ring-finger protein 125 (RNF125), carboxyl terminal of Hsp70-interacting protein (CHIP), Siglec-G/c-Cbl, and RNF122, lead to its degradation via a proteasome-dependent pathway, thus inhibiting RLR signaling [52,53,54,55]. Du et al. [56] identified a novel negative regulator of RLR-mediated IFN-I signaling, the leucine-rich repeat containing protein 25 (LRRC25). LRRC25 binds to RIG-I, which was modified by ISG15 (an ISG product), thus facilitating the interaction between RIG-I and the autophagic cargo receptor p62, which meditates the selective autophagy of RIG-I and downregulates IFN-I production (Figure 2). MAVS is located in mitochondria, functioning as a pivotal molecule in RLR signaling and as a critical target for RLR signaling pathway regulation. Ding et al. [57] showed that human parainfluenza virus type 3 (HPIV3) could induce mitophagy, thereby degrading MAVS and interfering with the RIG-I signaling-mediated IFN-I responses. Jounai et al. [58] confirmed that the ATG5–ATG12 conjugate downregulates the IFN-I yield by binding the RIG-I and MAVS CARD, showing that this conjugate is a crucial autophagy regulator. Jin et al. [59] demonstrated a new regulatory function of Beclin-1 in blocking the RIG-I–MAVS interaction. Beclin-1 can also interact with the MAVS CARD to inhibit RIG-I signaling. This research group also reported that Tetherin—an ISG product—is a negative regulator of RLR signaling, that functions by promoting MAVS degradation. Furthermore, the E3 ubiquitin ligase MARCH 8, can be recruited by Tetherin to catalyze K27-linked ubiquitin chains on MAVS at lysine 7, which initiates the NDP52-mediated selective autophagic degradation [60] (Figure 2). Autophagy has also been suggested to suppress RLR-mediated IFN-I signaling by targeting MAVS. ATG5 absence blocks the process of cellular autophagy, resulting in the accumulation of dysfunctional mitochondria, MAVS, and mitochondrial ROS. This process augments RLR-mediated IFN-I signaling, thus implying that autophagy’s ability to eradicate dysfunctional mitochondria is required for the balance of RLR-mediated IFN-I signaling [61]. IRF, another essential signal molecule, is also a regulatory target. Kim et al. [62] reported that the autophagy inhibitor, Rubicon, is involved in downregulating RLR signaling-mediated antiviral responses, which can suppress IRF3 dimerization by interacting with the IRF association domain, thereby downregulating IFN-I production (Figure 2).

These results provide essential evidence that clarifies how virus-induced autophagy inhibits host defenses. Studies by Ke et al. [63] and Shubham et al. [64] reported on the phenomenon that HCV-mediated autophagy could suppress RIG-I signaling and IFN-I production, which also occurred during the Dengue virus (DEV) infection [63]. They did not reveal the detailed mechanisms but showed that HCV-mediated autophagy upregulates the ATG5 and Beclin-1 expressions, which are critical for IFN-I production inhibition. Combined with the reports on the negative effects of ATG5–ATG12 and Beclin-1 in RIG-I signaling, it is not difficult to speculate some parts of the HCV-mediated autophagy mechanism that impair IFN responses.

### 3.2. Autophagy Regulates IFN-I Production via the TLR Signaling Pathway

The TLR signaling pathway is another important pathway that contributes to IFN-I production. TLR 3, 7, 8, and 9 are located in cytoplasmic endosomes, which sense virus RNA and unmethylated CpG DNA. All TLRs share a common structure comprised of an ectodomain that is a multiple leucine-rich repeat (LRR) and fulfills a function in sensing PAMPs; a transmembrane region; and a cytoplasmic tail with the Toll-IL-1R resistance (TIR) signaling domain that recruits adaptor proteins such as myeloid differentiation factor 88 (MyD88), MyD88-adaptor-like protein, TIR-domain-containing adaptor inducing interferon-β (TRIF), TRIF-related adaptor molecule, etc. Following the pathway activation, these adaptor proteins recruit a series of downstream signaling molecules, leading to IRF dimerization and IFN-I production [65].

Some mechanisms have been proposed by which autophagy regulates TLR signaling. During viral RNA recognition by TLR7, autophagy plays an essential role in the transmission of the viral replication to lysosomes and of IFN-α generation in plasmacytoid dendritic cells (pDCs) [12]. Frenz et al. [66] further confirmed this point by suggesting that autophagosomes capture the vesicular stomatitis virus (VSV) RNA and deliver it to endosomes, thereby facilitating IFN-I responses. During human immunodeficiency virus (HIV)-1 infections, pDCs mediate IFN-α production via TLR7 signaling, which also depends on the autophagy pathway [67]. LC3-associated phagocytosis (LAP) is a form of noncanonical autophagy contributing to the TLR9 signaling pathway [68]. According to Hayashi et al., after the TLR9 activation by CpG DNA, the autophagy protein LC3 and the inhibitor of nuclear factor kappa-B kinase α (IKKα) were recruited to endosomes where they formed the LC3-IKKα complex—a critical downstream driver of IRF7-mediated IFN-I production [69]. These studies revealed that the activity of TLR signaling pathway closely correlates with autophagy.

However, autophagy can also suppress TLR-triggered IFN-I production. Enterovirus 71 (EV71) and coxsackievirus A16 (CA16) can inhibit the TLR7-mediated IFN-I signaling pathway by accelerating endosome degradation via autophagy [70]. Moreover, selective autophagy plays an important role in balancing the TLR signaling. During polyinosinic-polycytidylic acid stimulation, autophagic cargo receptor NDP52 mediates the IRF3 inactivation by targeting adopter proteins TRIF and TRAF for selective degradation. Under normal circumstances, this autophagic process is restrained by ubiquitin-editing enzyme A20 [71]. Tripartite motif-containing (TRIM) 32, a vital immunoregulatory cytokine, has also been confirmed as a suppressor of the TLR3/4-mediated IFN-I responses by targeting TRIF, depending on TAX1BP1-mediated selective autophagy [72] (Figure 2).

Although some viruses can induce autophagy to elude immune defenses, the dominant role of autophagy in TLR signaling remains as the facilitation of viral recognition, which represents the antiviral strategy of autophagy independently of degradation. Furthermore, the positive regulation of autophagy in innate immunity could be used for clinical therapies designed to treat certain viral infections.

### 3.3. Autophagy Regulates IFN-I Production via the cGAS-Stimulator of Interferon Gene (STING) Signaling Pathway

The cGAS-STING signaling pathway is critical for transmitting DNA virus signals. In 2013, Gao et al. [73] first reported that cGAS possessed the ability to recognize HIV DNA. In normal cells, the activity of cGAS is suppressed by the monoglutamylation of cGAS, which is mediated by enzyme TTLL6. However, cytosolic carboxypeptidase5 (CCP5) and CCP6 could hydrolyze the monoglutamylation and polyglutamylation of cGAS, respectively, thus restoring cGAS synthase function [74]. Upon viral DNA stimulation, cGAS catalyzes the synthesis of the second messenger, cyclic guanosine monophosphate-adenosine monophosphate (cGAMP), which phosphorylates and dimerizes the key regulator, STING. STING recruits downstream TBK1 that phosphorylates and activates IRF3 to induce the expression of IFN-I [75,76].

The involvement of ATG proteins in regulating the cGAS-STING signaling pathway has been reported recently. In herpes simplex virus (HSV)-1 infected cells, Beclin-1 suppresses the DNA stimulation by interacting with cGAS, impeding the cGAMP synthesis and subsequent IFN-I production. Moreover, autophagy-mediated degradation can clear cytosolic pathogen DNA detected by DNA sensors, thus indirectly attenuating the cGAS-STING signaling pathway [77]. ATG9A is another negative regulator in balancing dsDNA-mediated immune responses, which suppresses the assembly of STING and TBK 1 [78]. Selective autophagy of the signaling molecules is also considered an important approach for regulating the cGAS-STING pathway. cGAS can be ubiquitinated through K63 linkage and recognized by autophagy cargo receptor p62, which directs specific autophagosomal degradations, leading to cGAS-STING signaling inhibition [79]. Persistent activation of the cGAS-STING signaling pathway leads to the elimination of STING, which is critical for immune homeostasis. Prabakaran et al. [80] showed that p62 could mediate the selective autophagy of active STING. The K63-linked ubiquitination of STING can be detected by p62, directing STING to the autophagic process, thereby avoiding cGAS-STING signaling overactivation and subsequent excessive IFN-I responses. Kimura et al. [81] also identified another selective autophagic receptor, TRIM21, which mediates IRF3 dimers autophagic degradation and inhibits cGAS-STING pathway-triggered IFN-I responses (Figure 2).

How autophagy and its molecular components interplay with PRR signaling is an important question for understanding the PRR signaling regulation [82]. Current studies have strikingly improved our understanding of this field and have identified two major mechanisms: the immunoregulatory function mediated by autophagy molecules and the degradation of signaling molecules mediated by selective autophagy. Collectively, autophagy constructs an elaborate and flexible system for balancing the innate antiviral immune responses, which largely enriches the existing regulatory network of PRR signaling.

## 4. Autophagy Mediates the Degradation of IFN Receptor

Following the activation of the above pathways, IFN-I is synthesized and released from the virus-infected cell in a paracrine or autocrine manner. Subsequently, IFN-I binds to the interferon-α/β receptor (IFNAR)—composed of two subunits IFNAR1 and IFNAR2—which stimulates downstream IFN-I pathways such as the Janus kinase (JAK)/signal transducer and activator of the transcription (STAT) pathway, thus inducing the expression of ISGs. The activity of IFN-I pathways is somewhat determined by the levels of IFNAR on the cell surface, indicating that the number of IFN receptors is essential for IFN-I responses during innate antiviral immunity.

Many viruses have developed strategies to promote the degradation of IFN receptors, thereby inhibiting IFN-I antiviral responses [83]. At this stage, endocytosis and lysosomal degradation mediated by Lys48- and Lys63-linked ubiquitination are considered the major pathways regulating the IFNAR levels [84,85]. However, recent evidence on HCV infections suggests a relation between autophagy and IFNAR degradation. Gunduz et al. [86] found that intracellular lipid accumulation induced the endoplasmic reticulum (ER) stress response and downregulated IFNAR1 levels in HCV infected Huh-7.5 cells that were cultured with free fatty acids (FFA). This resulted in the inhibition of the JAK-STAT pathway. Further research confirmed that HCV and FFA-induced autophagy promoted the expression of LAMP2A and the interaction between IFNAR1 and LAMP2A, indicating that CMA is responsible for mediating the IFNAR1 degradation [87]. Chandra et al. [88] also demonstrated that the HCV-mediated ER stress and autophagy selectively downregulated the IFNAR1 expression rather than the receptor of IFN-II and IFN-III, which is crucial for explaining the mechanisms’ HCV resistance to IFN-α and ribavirin. These studies together not only elucidate the correlation between CMA and IFNAR expression regulation but also offer a mechanism by which the HCV infection blocks IFN-I signaling selectively rather than IFN-III signaling. Most importantly, these results indicate that IFN-III possesses a stronger antiviral activity than IFN-I in inhibiting an HCV infection and that the suppression of HCV-induced ER stress and CMA is a potential strategy for reducing HCV resistance to IFN-I.

## 5. IFN-I System Regulates Autophagy

During innate antiviral immunity, IFN-I serves an immune-modulatory role rather than a direct antiviral role and constructs an antiviral defense by inducing numerous downstream ISG products. However, some studies have shown that IFN-α/β induces the autophagic degradation of viral components and promotes the delivery of PAMPs from the cytoplasm to endosomes containing TLR3. This indicates that, in addition to inducing ISG production, IFN-α/β uses autophagy against a virus [89,90]. It has also been demonstrated that the antiviral mechanisms of several ISG products have a relationship with autophagy. All of the above phenomena prove that autophagy can regulate antiviral IFN-I responses and that IFN-I and ISG products can eliminate viruses by mediating autophagy, indicating that the regulation between autophagy and IFN-I is a mutual process.

### 5.1. IFN-I Signaling Pathways Regulate Autophagy

Principal IFN-I signaling pathways include the JAK-STAT, phosphoinositide 3 kinase (PI3K)/protein kinase B (Akt)/mTOR, and mitogen-activated protein kinase (MAPK) pathways, which drive downstream ISGs expression and are crucial for antiviral immunity. Previous studies have shown that these pathways can regulate the autophagic process during starvation, inflammation, and antitumor immunity [91,92,93]. In 2011, via a high-content, image-based, genome-wide siRNA screen, Orvedahl et al. [94] identified 141 candidate genes that were required for viral autophagy, including several genes of the IFN-α signaling pathway. This study demonstrates that IFN-I signaling pathways may also be involved in antiviral autophagy. However, there have only been a few reports to date regarding the function of the IFN-I signaling pathways in regulating virus-induced autophagy. Here, we provide a summary of the IFN-I signaling pathways’ involvement in the regulation of autophagy induced by diverse factors.

The JAK-STAT pathway is a typical signaling pathway mediated by IFN-I. Following the binding of IFN-I to IFNAR, JAKs (TYK2 and JAK1) are activated via phosphorylation and then recruit and phosphorylate STAT1 and STAT2. This phosphorylation is required for STAT1 and STAT2 dimerization. The dimer binds to IRF9 and generates a complex called ISGF3, which translocates into the nucleus and interacts with the IFN stimulated response element (ISRE), leading to the ISGs’ transcription. Current studies have indicated that IFN-I can induce autophagy in multiple cancer cell lines and that several molecules of the JAK-STAT pathway are involved in the process [95,96,97,98]. In Daudi B cells, IFN-I triggers autophagy in a STAT2-dependent manner [95]. Zhu et al. [96] found that IFN-α induced autophagy in chronic myeloid leukemia cells and that the activation of JAK1 and STAT1 facilitated the generation of Beclin-1, a critical ATG protein for the PI3KC3 complex formation. IFN-β also induces autophagy in MCF-7 breast cancer cells in an ATG7- and STAT1-dependent manner [97].

The p38 MAPK pathway is another important pathway mediated by IFN-I. The activation of JAK1 leads to VAV protein phosphorylation and activation, which in turn activates the downstream regulator Rac1. Rac1 phosphorylates and activates the MAPK-kinase kinase (MAPKKK). Subsequently, MAPK kinase (MAPKK) is activated, resulting in the activation of MAPK [99]. MAPK is a highly conserved serine/threonine kinase, and there are at least six subfamilies described to date, including c-Jun N-terminal kinase (JNK)1/2/3, extracellular signal-related kinases (ERK)1/2, ERK3/4, ERK5/BMK1, ERK7/8, and p38MAPK (p38α/β/γ/δ) [100]. He et al. [92] showed that p38MAPK could enhance the inflammatory responses in microglial cells by phosphorylating ULK1 and inhibiting autophagy. Furthermore, JNK was also found to be involved in the autophagy of cancer cells by several mechanisms including upregulating the ATG5 expression, facilitating the Beclin-1 production, and triggering the dissociation of the Bcl-2/Beclin-1 complex [101,102,103]. Puissant et al. [104] confirmed that resveratrol (RSV) enhanced the autophagic cell death of chronic myelogenous leukemia cells by elevating the p62 yields in a JNK-dependent manner. Liu et al. [105] also demonstrated that palmitate (PA) induced apoptosis in bone marrow mesenchymal stem cells by activating the JNK and p38MAPK. In conclusion, these studies provide vital evidence for the crucial role of the MAPK pathway in inducing autophagy.

IFN-I can also trigger the PI3K/Akt/mTOR pathway. mTORC1 exerts a direct negative regulatory role on autophagy induction, which is a common regulatory target of the upstream autophagy activating signals. It has been shown that IFN-I initiates autophagy via the PI3K/Akt/mTOR pathway in cancer cells. Li et al. [98] found that autophagy induced by IFN-β was dependent on the PI3K/Akt/mTOR and ERK 1/2 signaling pathways in human glioma cells. IFN-I was also shown to decrease the mTORC1 activity in Daudi B cells and further limit the PI3K/Akt/mTOR pathway, resulting in the initiation of autophagy [95].

The JAK-STAT, PI3K/Akt/mTOR, and MAPK signaling pathways are crucial antiviral pathways. Given that numerous studies have identified the function of these pathways in the initiation of autophagy in cancer and inflammation, IFN-I triggered autophagy in anti-HCV immunity has been confirmed [90]. Orvedahl et al. also demonstrated that key molecules of the IFN-I signaling pathway were involved in viral autophagy [94]. The role of IFN-I and its downstream signaling pathways in inducing autophagy during a viral infection is concerning. Currently, research into antitumor tactics has benefited from the studies of IFN-I-induced autophagy in cancer cells [96,97]. Likewise, it is helpful for exploiting new antiviral targets to further elucidate the association of the IFN-I signaling pathways and autophagy during viral infections.

### 5.2. ISG Products Regulate Autophagy

ISG products are direct antiviral proteins and vital immune regulators. It has been reported that ISG products regulate and manipulate autophagy to exert their function in innate immunity. Its mechanisms can be summarized by two approaches: first, ISG products mediate cellular antiviral autophagy or prevent autophagy from being hijacked by the virus, and second, ISG products can trigger selective autophagy by targeting key molecules of antiviral immunity, thus regulating the IFN-I-induced antiviral effects.

Recently, considerable advances have been made in correlating ISG products’ antiviral effects and autophagy. Tallóczy et al. [8] demonstrated that RNA-dependent eIF2α protein kinase (PKR)—an important ISG product—was essential for the autophagic degradation of HSV-1. IFN-β-inducible protein SCOTIN has also been reported to impede the HCV infection by mediating the autophagic degradation of the HCV nonstructural 5A (NS5A) protein that exerts a vital role in the HCV replication [106]. It is known that certain viruses can manipulate autophagy to promote their multiplication. PML (also called TRIM19) can be induced by IFN-α/β/γ, which inhibits EV71-mediated autophagy to control viral multiplication [107,108]. Basler et al. [109] also found that paramyxovirus multiplication was restrained by Tudor domain containing 7 (TDRD7). TDRD7 possesses the ability to block paramyxovirus-triggered autophagy by interfering with the activity of adenosine 5’-monophosphate (AMP)-activated protein kinase (AMPK), which can activate ULK1 and suppress mTORC1 to induce autophagy [110,111]. Another ISG product, ribonuclease L (RNase L), can degrade and produce small viral RNA fragments, which activates autophagy to suppress viral multiplication during early viral infections [112]. Siddiqui et al. [113] independently revealed that RNase L initiated antiviral autophagy by mediating the JNK and PKR activation. Interestingly, both studies reported a similar phenomenon where the viral replication could be restrained by RNase L-induced autophagy in early infections, but in later stages, autophagy promoted the multiplication of the virus itself. Numerous studies have been published on the role of autophagy in promoting the replication of certain viruses. Therefore, the above phenomenon may be due to viruses hijacking autophagy for their reproduction during late infection stages, which counteracts and eclipses the RNase L-induced antiviral autophagy.

Selective autophagy is an essential approach for the immunoregulatory role of autophagy, and studies have shown that ISG products could mediate the selective autophagy of IFN-I responses to regulate antiviral immunity. The generation of TRIM14 in response to IFN-I, USP14, can be recruited by TRIM14 to remove the K48-linked ubiquitin chains of cGAS, thereby inhibiting the p62-mediated selective autophagy of cGAS and maintaining the activation of the IFN-I production pathways [79]. Another IFN-induced protein, ISG15 ubiquitin-like modifier (ISG15), can conjugate with target molecules, a process called ISGylation, and has been identified as a negative regulator for the RLR pathway by mediating the ISGylation of RIG-I [114]. Xu et al. [115] also found that IFN-I facilitated the ISGylation of Beclin-1 and further suppressed the formation of PI3KC3, thus blocking autophagy that may be subverted and utilized by viruses. Recent evidence has revealed a novel function of ISGylation. Similar to ubiquitination, ISGylation can trigger autophagic degradation. Nakashima et al. [116] demonstrated that ISG15 could interact with histone deacetylase 6 (HDAC6) and SQSTM1/p62, contributing to the autophagic degradation of ISG15-conjugated proteins. This was confirmed by another stud, where ISG15 was involved in the LRRC25-induced degradation of RIG-I via p62-dependent selective autophagy [56]. In this process, the LRRC25 only recognizes the RIG-I that was modified by ISG15.

## 6. Conclusions

The interplay between autophagy and innate antiviral immunity has grown to be an active area of investigation. An increasing number of studies have indicated that autophagy might be integrated into innate antiviral immunity where IFN-I responses serve as a vital bridge. Autophagy is involved in innate antiviral immunity via regulating the IFN-I system, and in turn, IFN-I and its induced proteins can also harness autophagy to achieve viral clearance, which enhances the ability of the immune system to respond to and combat viral invasions (Figure 3). The crosstalk between autophagy and IFN-I responses expands the antiviral immune regulatory network and benefits immune homeostasis, and our knowledge in this field will provide new therapeutic avenues for a wide range of viral- and immune-related diseases. However, the aspect of the IFN-I signaling pathway that regulates autophagy during viral infections remains to be elucidated, and the association between the antiviral activity of ISG products and autophagy also awaits further research.

## Figures and Tables

**Figure 1 viruses-11-00132-f001:**
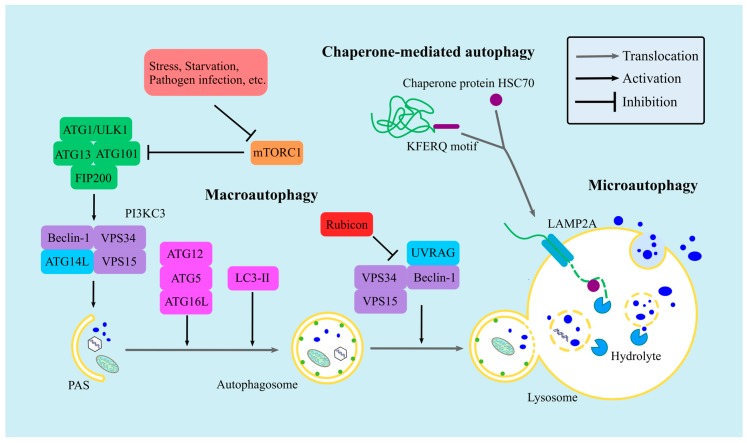
The types of autophagy. There are three autophagic routes: chaperone-mediated autophagy (CMA), microautophagy, and macroautophagy. In CMA, the KEFRQ motif of target proteins is recognized and captured by the chaperone protein HSC70. The complexes are then transported into the lysosome via a lysosomal receptor, LAMP2A. In microautophagy, cytosolic contents are sequestered by invagination of the lysosomal membrane, separated, and degraded in the lysosome. Macroautophagy is the dominant autophagic type, which is strictly directed by mTORC1 and several autophagy-related gene (ATG) protein complexes.

**Figure 2 viruses-11-00132-f002:**
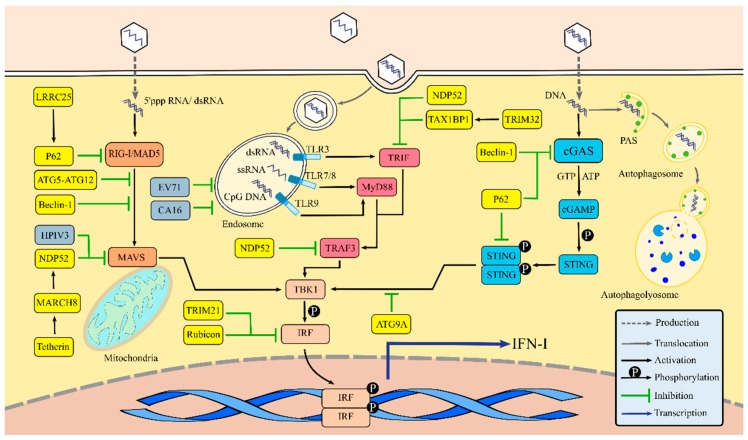
Autophagy regulates the expression of IFN-I. There are three major pattern recognition receptor (PRR) signaling pathways for regulating IFN-I expression, including the RLRs, TLRs, and cGAS-STING pathways. After recognizing a viral pathogen-associated molecular pattern (PAMP), PRRs recruit downstream signaling molecules, activate signaling cascades via TBK1 and IRF, and ultimately result in the expression of IFN-I. Autophagy can regulate virus detection by degrading viral PAMPs or delivering viral PAMPs to endosomes. Autophagy-related gene (ATG) proteins such as ATG5, ATG9A, ATG12, and Beclin-1 function as regulators of IFN-I production. Selective autophagy mediated by P62, NDP52, and TAX1BP1 plays a vital role in balancing the PRR-mediated IFN-I signaling. Viruses such as HPIV3, EV71, and CA16 can also manipulate key molecules of the autophagic process to inhibit IFN-I production.

**Figure 3 viruses-11-00132-f003:**
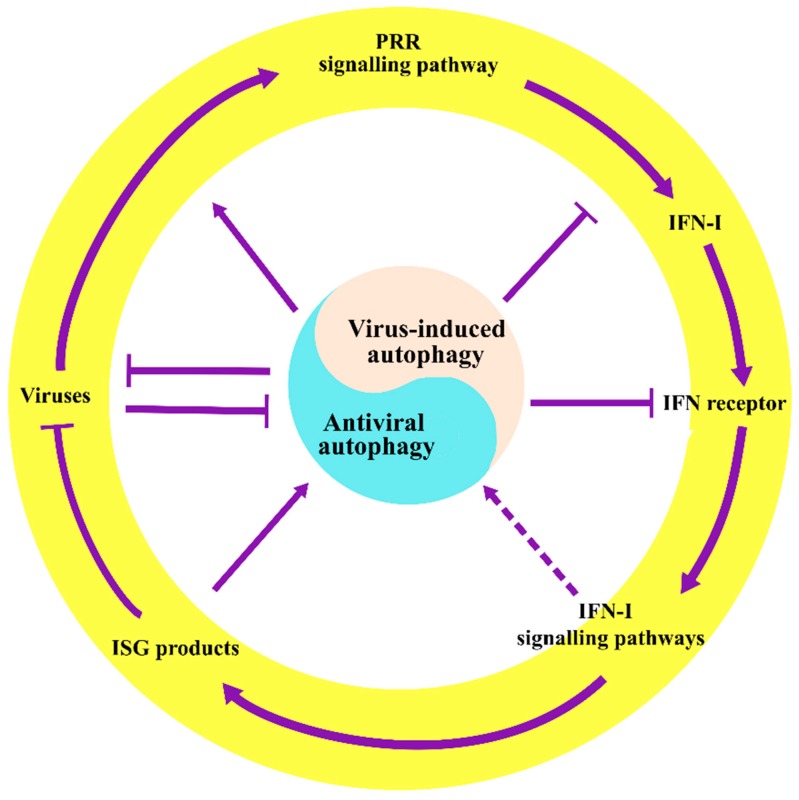
The crosstalk between autophagy and IFN-I-mediated antiviral immunity. Autophagy plays a vital role in regulating innate antiviral immunity induced by IFN-I. Virus-induced autophagy can suppress the IFN-I antiviral responses, and the IFN-I system can also manipulate autophagy to eliminate viruses. The crosstalk between autophagy and IFN-I responses may serve as a major bridge linking autophagy to innate antiviral immunity. Solid arrows indicate stimulation, T arrows indicate inhibition, and dashed arrows indicate potential stimulation.

**Table 1 viruses-11-00132-t001:** The targets of virus-IFN-I-response interactions via autophagy. Autophagy as a pro-viral or antiviral tool is mediated by viruses and IFN-I responses, respectively, the molecular mechanisms of which are intricate. Some of the targets are summarized in Table 1.

Viruses	Viral Targets for Inhibiting IFN-I Responses via Autophagy	Targets or Mechanisms of IFN-I Responses for Defending Against Viruses via Autophagy	References
HCV	RIG-I signaling		[63,64]
	IFNAR		[86,87,88]
		The degradation of HCV NS5A mediated by SCOTIN	[106]
		The degradation of NS3 mediated by IFN-β	[90]
DEV	RIG-I signaling		[63]
HPIV3	MAVS		[57]
VSV		PAMPs	[12,66]
HIV-1		TLR7 signaling	[67]
EV71	Endosome and TLR7 signaling		[70]
		The inhibition of EV71-mediated autophagy by PML	[108]
CA16	Endosome and TLR7 signaling		[70]
HSV-1	cGAS		[77]
		The autophagic degradation of HSV-1 mediated by PKR	[8]
Paramyxovirus		The inhibition of SeV-mediated autophagy by TDRD7	[14]

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
