# Peer review of "Crosstalk between Autophagy and Type I Interferon Responses in Innate Antiviral Immunity"

_viruses, 2019, doi:10.3390/v11020132_

Reviewer 1 Report

Although a potentially interesting topic for review, its value is diminished as few concrete conclusions are drawn. In fact, the authors conclude that the processes that regulate the proposed crosstalk between the type I IFN pathway and autophagy are largely unknown (ln 389-390). It is surmised that autophagy not only degrades viral components but also cellular factors, including components of antiviral immunity (ln 382-384), so that, the overall consequence to viral infection isn’t clear. In some instances, such as the section headed ‘Autophagy mediates the degradation of IFN receptor’ there is little evidence presented to support the authors assertion.

Figure 1 is useful as it summarises much of the literature and a greater focus on this analysis would improve the review. A greater focus on the consequence of major reports (such at Orvedahl ety al Nature 2011) and the consequence for the interferon pathway would better focus the topic. Also, a greater explanation of the different autophagy factors and their separate roles, with more detail on their involvement in specific cellular responses could clarify the topic. Overall the review lacks analysis and synthesis of the topic. Some concepts, such as ‘antiviral autophagy’ are not well justified and appear contradictory. As an instance, Figure 2 indicates 'antiviral autophagy' blocks the type I response, contradicting their own premise.

The document requires editing as the wording is often awkward and there are frequent errors of English (too numerous to list). Abbreviations are not explained. The early part of the review focuses excessively on review articles, rather than the primary literature. Although primary literature is cited in later sections, this concentrates on more recent reports and ignores important earlier publications. Also, there is little analysis of reported findings. No distinction is made between major reports and more minor reports. Instead, the findings are merely listed. There is also little attempt to reconcile apparent contradicting reports and no attempt is made to arrive at a compromise or speculate at reasons for apparent inconsistencies. No attempt is made to discern the overall weight of this pathway for the antiviral response.

Author Response

Point 1: Although a potentially interesting topic for review, its value is diminished as few concrete conclusions are drawn. In fact, the authors conclude that the processes that regulate the proposed crosstalk between the type I IFN pathway and autophagy are largely unknown (ln 389-390). It is surmised that autophagy not only degrades viral components but also cellular factors, including components of antiviral immunity (ln 382-384), so that, the overall consequence to viral infection isn’t clear.

Response 1: We appreciate the reviewer’s evaluation for our manuscript. The contents of type I interferon (IFN-I) responses should include IFN-I production pathways, IFN-I signalling pathways, and interferon-stimulated gene (ISG) responses. It is true that the interplay between IFN-I signalling and autophagy pathways still require more exploration, but which may not represent the crosstalk between the complete IFN-I responses and autophagy. Meanwhile, we believe that it is also meaningful to point out the part of the field that deserves further research.

     In viral infection, autophagy may play a pro-viral or anti-viral role, and the precise role varies for different viruses largely. It may be inappropriate to conclude the explicit function of the autophagy in the viral infection from an overall perspective. Also, there have been lots of reviews on autophagy and virus, our purpose in initiating this review is to discuss the association between autophagy and antiviral immunity by summarizing the crosstalk of IFN-I response and autophagy.

     We have re-written the section of conclusions to make our purpose clearer.

Point 2: Figure 1 is useful as it summarises much of the literature and a greater focus on this analysis would improve the review. A greater focus on the consequence of major reports (such as Orvedahl et al Nature 2011) and the consequence for the interferon pathway would better focus the topic.

Response 2: We appreciate and agree with the reviewer’s advice.

We have made improvements in adding the analysis for the important reports in the revised manuscript.

We also strongly agree that the report of Orvedahl et al. is essential and valuable for proving the correlation between virus-induced autophagy and interferon (IFN) pathways. However, there are lots of reports in the field of interaction between IFN pathways and autophagy induced by cell stress, cancer, inflammation and so on, but little in the virus-induced autophagy. Given that current studies have been identified, we can only speculate that the interplay between IFN pathways and virus-induced autophagy is a potential valuable area and requires more works.

For the part of the IFN pathways’ consequence, we have made a detailed summary in section 5.2.

Point 3: In some instances, such as the section headed ‘Autophagy mediates the degradation of IFN receptor’ there is little evidence presented to support the authors assertion.

Response 3: We have noticed that some of the content in this section may be inaccurate, and have deleted it in the revised manuscript.  However, we respectfully disagree this section is wrong, and some related references are as follows:

Gunduz, F.; Aboulnasr, F.M.; Chandra, P.K.; Hazari, S.; Poat, B.; Baker, D.P.; Balart, L.A.; Dash, S. Free fatty acids induce ER stress and block antiviral activity of interferon alpha against hepatitis C virus in cell culture. Virol J 2012, 9, 143, doi:10.1186/1743-422X-9-143.

Kurt, R.; Chandra, P.K.; Aboulnasr, F.; Panigrahi, R.; Ferraris, P.; Aydin, Y.; Reiss, K.; Wu, T.; Balart, L.A.; Dash, S. Chaperone-Mediated Autophagy Targets IFNAR1 for Lysosomal Degradation in Free Fatty Acid Treated HCV Cell Culture. PLoS One 2015, 10, e0125962, doi:10.1371/journal.pone.0125962.

Chandra, P.K.; Bao, L.; Song, K.; Aboulnasr, F.M.; Baker, D.P.; Shores, N.; Wimley, W.C.; Liu, S.; Hagedorn, C.H.; Fuchs, S.Y., et al. HCV infection selectively impairs type I but not type III IFN signaling. Am J Pathol 2014, 184, 214-229, doi:10.1016/j.ajpath.2013.10.005.

So it is appropriate to keep this section in our manuscript, we have re-written this section in the revised manuscript. Many thanks for the reviewer’ advice.

Point 4: Also, a greater explanation of the different autophagy factors and their separate roles, with more detail on their involvement in specific cellular responses could clarify the topic.

Response 4: We appreciate the reviewer’s advice. We have drawn a new figure to display the role of different autophagy-related gene (ATG) proteins in autophagic degradation. More contents also have been added and fixed to explain the function of autophagy factors that were mentioned in the manuscript.

Point 5: Overall the review lacks analysis and synthesis of the topic. Some concepts, such as ‘antiviral autophagy’ are not well justified and appear contradictory. As an instance, Figure 2 indicates 'antiviral autophagy' blocks the type I response, contradicting their own premise.

Response 5: We appreciate that the reviewer points out our deficiency. We have added more analysis and discussion for those significant original reports in our revised manuscript. The disputable concept antiviral autophagy’, we have fixed it in the article and checked other concepts that may be controversial. Many thanks for the reviewer’s suggestions.

Pointed 6: The document requires editing as the wording is often awkward and there are frequent errors of English (too numerous to list). Abbreviations are not explained.

Response 6: We apologize for the language problems in the original manuscript and carefully checked and revised the entire manuscript for grammatical and formatting errors. We also added the full name of those abbreviations that are not explained.

Pointed 7: The early part of the review focuses excessively on review articles, rather than the primary literature. Although primary literature is cited in later sections, this concentrates on more recent reports and ignores important earlier publications.

Response 7: According to the reviewer’s advice, we have added original articles as references, which are important for understanding the early part. In fact, this is a mistake of us. Many thanks for the reviewer’s advice.

Pointed 8: Also, there is little analysis of reported findings. No distinction is made between major reports and more minor reports. Instead, the findings are merely listed. There is also little attempt to reconcile apparent contradicting reports and no attempt is made to arrive at a compromise or speculate at reasons for apparent inconsistencies. No attempt is made to discern the overall weight of this pathway for the antiviral response.

Response 8: We deeply appreciate the valuable advice from the reviewer. In the revised manuscript, we have added more discussion to present the significance of reports in the article. Many sections of the article have been reorganized to distinguish the major or minor reports. In addition, for those reports that may lead to conflicting results, more analysis has been added to clarify the reason. We also re-wrote the “Conclusions” and added more contents to elucidate values of the topic.

     The reviewer is worthy of respect, and we appreciate his/her review and suggestions for our manuscript.

Reviewer 2 Report

This is a very scholarly review that will be of significant interest to virologists.  I have only a few comments for the authors to consider

More visual representation , e.g. of the different subsets of autophagy, might be of value to those readers who are not experts in the field

A summary table listing the different viruses and there effects on autophagy might also be of value

Considering all that was well described in the manuscript, are there an differences to type 1 IFNs that involve Type III IFNs (interferon lambda). For example are there any reports suggesting that the Type III IFN receptor is more resistant to degradation?

Although a bit outside the scope of the review, are there any clinical trials that are being conducted based on what the authors have covered.  Please understand that I am not suggesting all IFN clinical trials be covered.  Rather are there any that are trying to exploit the biology covered in this review.

Author Response

Point 1: This is a very scholarly review that will be of significant interest to virologists.

Response 1: We appreciate the reviewer’s positive evaluation for our manuscript.

Point 2: More visual representation, e.g. of the different subsets of autophagy, might be of value to those readers who are not experts in the field.

Response 2: We agree with the reviewer’s advice. We added one figure in our revised manuscript, which displays the different forms of autophagy to readers who may not be familiar with this field.

Point 3: A summary table listing the different viruses and their effects on autophagy might also be of value.

Response 3: We have added a summary table in the revised manuscript. Considering that the subject of “different viruses and their effects on autophagy” usually appears on the papers that demonstrate the relationship between viruses and autophagy, the theme of our table focuses on the targets of interaction between the virus and IFN-I response via autophagy.

Point 4: Considering all that was well described in the manuscript, are there differences to type 1 IFNs that involve Type III IFNs (interferon lambda). For example, are there any reports suggesting that the Type III IFN receptor is more resistant to degradation?

Response 4: We deeply appreciate the reviewer’s inspiration and advice. We also noticed some literature that indicate the difference of type I interferon (IFN-I) receptor and Type III interferon (IFN-III) receptor in the virus-mediated autophagic degradation. We have discussed this vital point in our revised manuscript.

Point 5: Although a bit outside the scope of the review, are there any clinical trials that are being conducted based on what the authors have covered.  Please understand that I am not suggesting all IFN clinical trials be covered.  Rather are there any that are trying to exploit the biology covered in this review.

Response 5: We could not give a definitive answer on whether there are any applications or trials related to our manuscript have entered the clinical trial stage. But many studies that were cited in our manuscript possess the significance for guiding clinical treatment, and we are confident that autophagy will be exploited as a therapeutic target or tool in the near future. In the revised manuscript, we also added much analysis on the application potential of the researches that we summarized. 

The reviewer’s advice for our manuscript is valuable and highly appreciated.

Round  2

Reviewer 1 Report

The document is modesty improved. It requires editing by a professional english service before it can be published.

Author Response

Response to Reviewer 1 Comments

Point 1: The document is modesty improved. It requires editing by a professional english service before it can be published.

Response 1: We appreciate the reviewer’s evaluation for the revised manuscript. Language presentation has been improved with assistance from English editing service provided by MDPI.